

# Building healthy eating habits in childhood: a study of the attitudes, knowledge and dietary habits of schoolchildren in Malaysia

Kazi Enamul Hoque, Megat Ahmad Kamaluddin, Ahmad Zabidi Abdul Razak and Afiq Athari Abdul Wahid

Faculty of Education, University of Malaya, Kuala Lumpur, Malaysia

## ABSTRACT

**Background:** Overweight and obesity have increased rapidly in incidence to become a global issue today. Overweight and obesity problems are significantly linked to unhealthy dietary patterns, physical inactivity and misperception of body image. This study aimed to determine whether Malaysian children build healthy eating habits from childhood.

**Methods:** A survey on eating habits was conducted among primary school students in standards 4 to 6 in the state of Selangor, Malaysia. The findings of the study were reported in the form of descriptive statistics involving frequencies and percentages. Data from 400 respondents were analyzed.

**Results:** Our findings showed that the students understood the definition of healthy food and the types of food that are considered healthy. Although the students knew that food such as deep-fried drumsticks and hamburgers contain a high amount of saturated fat and cholesterol, these foods were still consumed by them. There was also a high consumption of foods that are fried and contain sugar, salt and saturated fat. In choosing food, two major factors contributed to the students' decisions: cleanliness (65.8%) and the preference of their parents (12.3%).

**Discussion:** Our findings indicate that by implementing the Integrated School Health Program (ISHP) properly, students' eating habits can be improved by creating a school with a healthy environment.

Corresponding author
Kazi Enamul Hoque,
tffr2011@yahoo.com

## INTRODUCTION

Obesity has increased rapidly in incidence to become a global issue today (*Kesten, Cameron & Griffiths, 2013*; *World Health Organization, 2013*). With the vast advancements of the modern world, food is developed in ways that are cheap for consumers but cause the food to contain high amounts of saturated fat and cholesterol (*Holliday et al., 2014*). However, not all fats cause obesity as evidences show that substitution of saturated fat with polyunsaturated fat improves cardiometabolic outcomes (*Imamura et al., 2016*). Apart from that, obesity is further exacerbated by the changing lifestyles of individuals,

including changes in methods of transportation and a lack of exercise due to work constraints. Obesity is also a risk factor for other non-communicable diseases and is becoming an increasing problem in both developing and developed countries (*Ellulu et al., 2014*; *World Health Organization, 1998*). Obesity is evident among both adults and the younger generation. Studies have shown that overweight and obesity problems are significantly linked to unhealthy dietary patterns, physical inactivity and misperceptions of body image (*Egner, Oza-Frank & Cunningham, 2014*; *Sandercock, Voss & Dye, 2010*; *Khor et al., 2009*; *Nagel et al., 2009*). Furthermore, according to a study conducted by the University Kebangsaan Malaysia (UKM), the percentage of obese children among primary school students in Malaysia has increased from 9.7% in 2001 to 13.7% in 2007 (*Karim, 2014*). The increase of 4% within six years is an alarming number, and according to *Birch (1999)*, the students involved will have the tendency to continue their food attitudes and eating habits throughout adulthood. Thus, it is important for schools to implement measures aimed at inculcating healthy eating habits as early as the primary school level to develop an ongoing healthy lifestyle. Additionally, the development of a healthy lifestyle is important because a healthy body enhances academic performance (*Brown et al., 2007*; *Egner, Oza-Frank & Cunningham, 2014*; *Giovannini, Agostoni & Shamir, 2010*; *Pearson, Biddle & Gorely, 2009*). Moreover, schools are uniquely positioned to shape children's eating habits (*Gourdet et al., 2014*). Thus, the possibility of correcting eating habits is possible through changing the eating habits from low-nutrient, high-energy diets to nutritious food and food with sufficient energy. This can be done through education and by creating an a awareness campaign. With this purpose, the Malaysian School Health Program began in 1967 with the main objective of developing students who are productive and healthy. In 2011, the Healthy Environment Cabinet Committee (JKPHS) Series No. 1 chaired by the Deputy Prime Minister of Malaysia, Tan Sri Dato' Haji Muhiyiddin bin Haji Mohd Yassin, had decided on strengthening healthy school environments through the implementation of efforts towards a healthy diet and by avoiding the availability of unhealthy food and drinks, in line with the implementation of *Program Bersepadu Sekolah Sihat (PBSS)* or the Integrated School Health Programme (ISHP) (*Ahmad Husairi, 2014*). In cooperation with the Ministry of Education and the Ministry of Health, the School Health Program was given the new name of the *PBSS,* or the ISHP. The purpose of the program was to synchronize all school activities that were related to health. Some activities that were introduced included the *Rancangan Makanan Tambahan (RMT),* or the Supplementary Food Program (SFP), which helps provide an adequate nutritional intake for students to ensure a well-balanced diet, and the *Program Susu Sekolah (PSS),* or the School Milk Program (SMP), which provides fresh milk for students to consume calcium. Other facilities, such as specific rooms for dental care and sick bays, were equipped in schools to aid in achieving the objectives of the ISHP. However, it had not been determined whether students are aware of ISHP objectives. Therefore, a study is necessary to explore students' awareness and the performance of the ISHP programs implemented by the schools.
## MATERIALS AND METHODS

This was a survey conducted to understand the eating habits of primary school students from standards 4–6 within the age range of 10–12. Data collection from a large number of students was necessary to generalize the results. There were approximately 554, 994 primary school students (standards 1–6) in Selangor District (*Department of Statistics, 2015*). The number of students would be fewer (not more than 350,000) from standards 4 to 6. In accordance with Gay, if the population is 50,000, a sample size of 1% would be more than adequate (*Gay, 1996*). *Krejcie & Morgan (1970)* also determined a sample size of 381 for a population of 50,000, 382 for a population of 75,000 and 384 for a population of 100,000. They also suggested a sample size of 1% is adequate if the population is more than 100,000. Thus, the sample size (400 students) for this study is more than adequate to truly represent the total population as total students from standards 4–6 will not be more than 350,000 in Selangor District of Malaysia.

### Participants

There are two types of public schools at the primary leve: the National Schools and the National Type Schools. In the National Schools, the medium of instruction is the Malay language, which is the national language. The medium of instruction in the National Type Schools is either the Chinese language or the Tamil language. In both types of schools, the Malay language is a compulsory subject. English is compulsory and is taught as a second language in all schools. Chinese, Tamil and indigenous languages are also offered as subjects in the National Schools. All three types of public schools are included in this study; these comprise Sekolah Kebangsaan (National Schools), Sekolah Kebangsaan Jenis Tamil (National Type Schools) and Sekolah Kebangsaan Jenis Cina (National Type Schools). Invitations were sent to the respective schools until the number of schools required was achieved. In response to the invitations, the principals were given the responsibility to address the matter with the teachers and students involved in the study. After seeking agreement from the school to participate in the study, a number of teachers involved in the subject of *Pendidikan Jasmani dan Kesihatan (PJK),* or Physical and Health Education, were asked to choose a class taught from either standards 4, 5 or 6 in order to obtain answers to the questionnaire.

### Instrumentation: validity and reliability

A structured bilingual version of the questionnaire, comprising Malay and English, was used for the study. A student questionnaire was constructed for the purpose of assessing students in standards 4, 5 and 6 on the current knowledge, attitudes and practices about healthy eating.

The questionnaire was adapted from the Hong Kong Department of Health (*Department of Health, 2009*), and a pilot study was conducted to either modify or omit questions to suit the context of Malaysian schools. A pilot study was conducted in February 2015 to assess the content validity and comprehensibility of the measurement tools and to test the logistics of the study. Apart from that, before the study the opinions of experts have been taken and the questionnaire was corrected based on their suggestions

to improve the validity. A total of 45 students from three primary schools participated in the pilot study. Over 15 students were invited from each school with the selection of five students from each Grade. These three schools were excluded from the final study. The final questionnaire was revised based on the feedback from the pilot study. The Cronbach's Alpha of the instrument was between 0.60–0.90. This pilot study shows that majority of the questions are valid and produce reliable results for analysis; thus, no change was needed for the actual survey.

## Procedure

As the primary school children are involved, permission has been taken from University of Malaya Ethics Committee. Between April 15, 2015 and April 19, 2015, 16 participating schools were identified, and the questionnaires were distributed to the students. The representatives involved in this study distributed the questionnaire to all the classes of standards 4, 5 and 6 that were chosen. All the students involved were asked to complete the questionnaire during a classroom session. The schools were given four weeks to conduct the questionnaires. The completed questionnaires were collected by the teachers. A representative performing the study collected all the questionnaires from all the schools involved. The completed questionnaires were collected from the schools between May 6, 2015 and June 8, 2015.

## Data analysis

Descriptive statistics is used for all data representation of all survey questions. Generally, descriptive statistics is the easiest form of summarizing data in a presentable format. This simple statistics is performed in Statistical Package for Social Science version 20.0. Some of the most common function used for this survey is frequency, percentage, standard deviation and mean.

## RESULTS

### Rate of responses

A total of 600 questionnaires were distributed by the teachers to the students in standards 4, 5 and 6 at the 16 participating schools. A total number of 420 answered questionnaires were returned, resulting in a 70% response rate. Twenty of these questionnaires were unanswered, and the remaining 400 were used in the analysis. Respondents' demographics Table 1 shows that the majority of respondents (40%) were students in standard 6. The respondents also included more male students (53.5%) and more students belonging to the Malay race (65.8%).

### Students' knowledge of healthier food and their preferences

In identifying knowledge about healthy food, six pairs of food were given in the questionnaire. The students were asked to choose the healthiest food among the six pairs. Table 2 indicates that the majority of students understood what healthy foods were based on the correct choices made.

**Table 1 Characteristics of 400 children aged 10–12 years in standards 4–6 in the state of Selangor, Malaysia.**

| Variables | No. of students | Percentage |
|---|---|---|
| Grade (n = 400) | | |
| Standard 4 | 143 | 35.8 |
| Standard 5 | 97 | 24.3 |
| Standard 6 | 160 | 40.0 |
| Gender (n = 400) | | |
| Male | 214 | 53.5 |
| Female | 186 | 46.5 |
| Race (n = 400) | | |
| Malay | 263 | 65.8 |
| Chinese | 44 | 11.0 |
| Indian | 93 | 23.3 |
| School district (n = 400) | | |
| Selangor | 400 | 100 |

**Table 2 Knowledge of healthier food and their preferences of 400 children aged 10–12 years in standards 4–6 in the state of Selangor, Malaysia.**

| Students' knowledge of healthier food choices | | | | | Students' food preferences | | | | |
|---|---|---|---|---|---|---|---|---|---|
| Healthier choice | % | Unhealthier choice | % | | Healthier choice | % | Unhealthier choice | % | |
| Soya sauce drumsticks | 55.0 | Deep fried drumsticks | 45.0 | | Soya sauce | 48.7 | Deep fried drumsticks | 51.3 | |
| Pure orange juice | 86.3 | Carbonated drinks | 13.7 | | Pure orange juice | 79.3 | Carbonated drinks | 20.7 | |
| Raisin whole-meal | 82.0 | Hot dog | 18.0 | | Raisin whole-meal | 52.3 | Hot dog | 47.7 | |
| Yogurt | 80.0 | Ice cream | 20.0 | | Yogurt | 51.5 | Ice cream | 48.5 | |
| Spaghetti with fresh tomato sauce | 77.3 | Hamburger and fries | 22.7 | | Spaghetti with fresh tomato sauce | 39.8 | Hamburger and fries | 60.2 | |
| Fried noodles with vegetables | 79.0 | Fried noodles with chicken | 21.0 | | Fried noodles with vegetables | 43.8 | Fried noodles with chicken | 56.2 | |

In identifying the students' food preferences, the same six pairs of food and drink as the knowledge questions were provided. Table 2 also shows that there was a difference in the answers between the students' preferences and their perceived opinions on what healthy foods are. The percentages of students' food preferences were generally lower. For example, although many students perceived that deep fried drumsticks, hamburgers and fries and fried noodles and chicken are unhealthy foods, more than half of the respondents still preferred eating those foods despite the fact that they are unhealthy choices.

## Reasons for not consuming breakfast

A total of 67%, or 268 students, had consumed breakfast on the day of the survey. The students who did not consume breakfast showed different reasons. Table 3 shows that an alarming figure of 56.8% of the students did not have breakfast because they did not have enough time, and 22.7% were not in the habit of having breakfast. Another concerning reason was that nobody had prepared breakfast for 7.6% of them.

**Table 3 Reasons for not having breakfast of 400 children aged 10–12 years in standards 4–6 in the state of Selangor, Malaysia.**

| Reason | No. of students | Percentage |
|---|---|---|
| I do not have time | 75 | 56.8 |
| I am on a diet | 2 | 1.5 |
| I want to save money | 0 | 0.0 |
| I am not used to having breakfast | 30 | 22.7 |
| I do not have the appetite to eat in the morning | 15 | 11.4 |
| No one has prepared breakfast for me in the morning | 10 | 7.6 |
| Other reasons | 0 | 0.0 |

**Table 4 Eating habits of various food categories on average every day of 400 children aged 10–12 years in standards 4–6 in the state of Selangor, Malaysia.**

Eating frequency

| Food category | Percentage of students | | | | | |
|---|---|---|---|---|---|---|
| | No. of students | More than twice | Twice | Once | Never | Do not know |
| Fruits | 400 | 47.0 | 18.8 | 26.0 | 5.8 | 2.4 |
| Vegetables | 400 | 44.5 | 24.0 | 21.0 | 6.8 | 3.7 |
| Dairy products | 400 | 30.3 | 18.5 | 31.8 | 14.8 | 4.6 |
| Meat, chicken, fish | 400 | 58.5 | 21.5 | 14.5 | 3.0 | 2.5 |
| Grains | 400 | 63.8 | 17.3 | 11.3 | 3.5 | 4.1 |
| Fried and deep-fried food | 400 | 37.0 | 23.8 | 28.8 | 7.3 | 3.1 |
| Drinks with added sugar | 400 | 15.3 | 20.3 | 40.0 | 20.5 | 3.9 |
| Food high in sugar | 400 | 20.2 | 16.2 | 40.1 | 16.0 | 7.5 |
| Food high in salt | 400 | 12.0 | 11.0 | 30.5 | 40.0 | 6.5 |
| Food high in fat | 400 | 24.5 | 17.5 | 39.8 | 11.8 | 6.4 |

## Students' daily dietary habits

The students were required to evaluate their daily dietary habits during the week prior to answering the questionnaire. According to Table 4, the percentage of students consuming fruits and vegetables more than twice was, on average, between 44.5 and 47.0%. However, there was a high occurrence of unhealthy eating habits, such as having deep-fried food and consuming foods high in sugar, salt and saturated fat. Between 60% and 92.7% of the students did not reach the recommended frequencies, as shown in Table 5.

## Students' knowledge of general healthy eating

Based on the two questions asked, there was a lack of healthy eating knowledge by the students because only 27.5% correctly chose grains as the type of food that should be provided most often in a healthy lunchbox and only 35.8% knew the correct recommended servings of fruits and vegetables (Table 6).

## Students' snack eating habits and sources

Students were shown pictures of snacks before answering whether they had the habit of eating them. The majority of students (73.3%) had the habit of snacking; they mostly

**Table 5 Percentage who reached the corresponding recommended frequency of 400 children aged 10–12 years in standards 4–6 in the state of Selangor, Malaysia.**

| Recommended eating frequency of various food categories | No. of students | No. of students/percentage of students | |
| --- | --- | --- | --- |
| | | Reached recommended frequency | Not reached recommended frequency |
| Fruit $\geq$ 2 times | 400 | 263 (65.8%) | 137 (34.2%) |
| Vegetables $\geq$ 2 times | 400 | 274 (68.5%) | 126 (31.5%) |
| Dairy products $\geq$ 1 time | 400 | 322 (80.5%) | 78 (19.5%) |
| Meat, chicken, fish $\geq$ 1 time | 400 | 320 (80.0%) | 80 (20.0%) |
| Grains $\geq$ 2 times | 400 | 255 (63.8%) | 145 (36.2%) |
| Not having fried and deep-fried food | 400 | 29 (7.3%) | 371 (92.7%) |
| Not having drinks with added sugar | 400 | 82 (20.5%) | 318 (79.5%) |
| Not having food high in sugar | 400 | 63 (15.8%) | 337 (84.2%) |
| Not having food high in salt | 400 | 160 (40.0%) | 240 (60.0%) |
| Not having food high in fat | 400 | 47 (11.8%) | 353 (88.2%) |

**Table 6 Percentage of students correctly answering the general healthy eating messages.**

| Healthy eating messages | No. of students | Percentage |
| --- | --- | --- |
| A healthy lunchbox should have more grains | 110 | 27.5 |
| A person must consume two servings of fruits and three servings of vegetables daily | 143 | 35.8 |

had homemade snacks (86.5%), while 2.3% bought snacks from the school co-operative shop. Others sources of snacks included outside shops and supermarkets (11.2%).

## Matters for the students' consideration in selecting their food

In the event of choosing food, students were particular regarding the cleanliness of the food: 65.8% responded to cleanliness as a high priority, followed by 12.3% responding that the preference of parents determined food choice, as shown in Table 7.

## Students' perceived eating habits

Figure 1 shows that a total of 45.5% of students viewed themselves as having very healthy eating habits, and only 1% considered their eating habits to be very unhealthy

## Students' awareness of the Integrated School Health Program (ISHP)

More than 45% of the students had heard of the ISHP, whereas 55% did not know about the program.

## Support in promoting healthy eating at schools

A total of 77.8% of the students stated that they supported healthy eating at school, whereas only 8.8% of the students were unsupportive of healthy eating. Another 13.4% did not know whether there was any program of promoting healthy eating at schools.

**Table 7 Consideration for choosing food of 400 children aged 10–12 years in standards 4–6 in the state of Selangor, Malaysia.**

| Matters for consideration | No. of students | Percentage |
| --- | --- | --- |
| Cleanliness | 263 | 65.8 |
| Taste | 23 | 5.7 |
| Good for health | 17 | 4.2 |
| Freshness | 43 | 10.7 |
| Price | 4 | 1.0 |
| Easy to get | 1 | 0.3 |
| Preference of parents | 49 | 12.3 |
| Choice of other schoolmates | 0 | 0.0 |

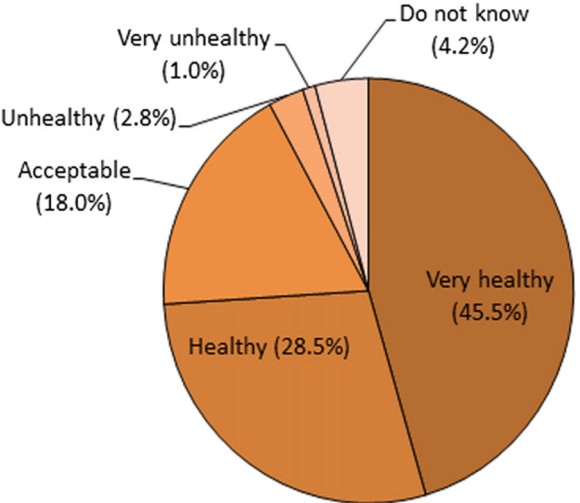

**Figure 1 Students' perception of their eating habits: data of 400 children aged 10–12 years in standards 4–6 in the state of Selangor, Malaysia.**

## Participation in school activities promoting healthy eating

Based on the students' responses regarding their participation in school activities that promoted healthy eating in the past year, only 13% of the students claimed to have been involved. Nearly half of the students (44.8%) claimed they were not sure of their participation in these types of activities, and 29.3% of the students reported that no such activities were ever organized in their schools.

## DISCUSSION

Thus, the ISHP is a program that is still considered important in promoting a healthy eating environment. It is of utmost importance for students to learn healthy eating habits at an early age, thus defining the purpose of implementing the program at the primary level (*Gourdet et al., 2014*). This survey provides an opportunity to identify the dietary patterns of students and the elements that affect their eating habits. Findings show that the students understand the definition of healthy food but still 50% of the students consume deep-fried drumsticks and hamburgers containing a high amount of

saturated fat and cholesterol. Approximately 60% of the students have knowledge of the correct intake of foods such as vegetables, fruits, meats and grains. At the same time, a high percentage of students consume foods that are fried and contain sugar, salt and fat. Parents play an important role in determining the types of snacks as 86.5% of students responded that the snacks they consumed were taken from their homes. Another finding of students not having breakfast (67%) may cause serious health concern in later life.

## Students' perceptions towards eating frequencies in various food categories

In terms of the students' practice in eating healthy foods, there was a high correct response rate regarding the recommendations of the eating frequency in the various food categories. The students responded well in terms of the correct intake of foods such as vegetables, fruits, meats and grains, with an average correct response rate of more than 60%. However, there were a high percentage of students who consumed foods that were fried and contain sugar, salt and saturated fat. This finding is in line with the high percentage of students preferring fried drumsticks in the question regarding the pairs of food. Based on these results, culture is an aspect that can be examined in terms of the types of food consumed. Malaysia is a country that has infused its food based on the different racial backgrounds. Most of the food that can be found and bought in Malaysia is prepared in ways that can be seen as unhealthy. This finding is supported by *Noraziah & Mohd Azlan (2012)*, who stated that current Malaysian foods are sweet, oily and fatty, which have quality in taste but does not consider the aspect of health. Furthermore, with parents spending more time at work, students are exposed to foods that are prepared quickly, such as microwaved food in boxes and eating at fast food restaurants, where food can easily be ordered and sent directly to homes. As a result of these factors, the types of food consumed contain ingredients that are high in sugar, salt and fat, leading to unhealthy eating behaviors.

## Problems students face in consuming breakfast

Breakfast is considered the most important meal of the day because it helps an individual to work and think during the early part of the day (*Pandey et al., 2013*). Without breakfast, students may not have the energy to think and perform tasks in class. According to the survey, only 67% of students had breakfast on the day of the survey. For those who did not have breakfast, 56.8% responded that not having enough time was their reason for missing breakfast. As such, *Singleton & Rhoads (1982)* also identified a lack of time (43%) as the main factor for not having breakfast. This effect may be due to the problem of time management. In most school systems, school sessions begin as early as 7:30 in the morning, thus requiring students to be in school before 7:30. Students must wake up early and prepare themselves for school. Students using transportation, such as school buses, must prepare themselves even earlier because the buses run early in the morning to pick up all the students in different areas to bring to school. As a result, some students have no time for breakfast and may wait until the morning recess to eat.

### Students' perceptions and action towards healthy food

According to the survey, the majority of students were able to correctly identify the six pairs of food with each pair having a minimum of 55% correct answers. This finding indicates that the students understood the definition of healthy food and the types of foods that are considered healthy. However, in a question involving those same food pairs and the students' choice of food, there was a difference in the percentage of correct choices. Although the students knew that foods such as deep-fried drumsticks and hamburgers contain a high amount of saturated fat and cholesterol, these food are still consumed by the students, with more than 50% of the survey population reporting consumption of these foods. Only having knowledge about healthy eating habits is not sufficient; the practical aspect should also be emphasized to students, because there is no purpose of learning without implementing what has been taught (*Woodruff & Hanning, 2009*).

### Students' habit of snacking

Snacks can be classified as foods that are eaten between meals or consumed as light meals and can be either healthy or unhealthy. Healthy snacks, such as fruits, vegetables and plain water, contribute to healthy eating and contain no saturated fat or artificial flavors. However, snacks such as chips and sweets contain an excess of ingredients, such as salt and sugar, that can lead to obesity. Moreover, several studies have indicated that the consumption of unhealthy snacks is one of the main contributors to being overweight (*de Graaf, 2006*). It was found that 73.3% of students have the habit of snacking, and 86.5% of students responded that the snacks they consumed were taken from their homes. This finding means that parents play an important role in determining the types of snacks that their children are eating. Parents should set an example for their children and differentiate between healthy and unhealthy food. This conclusion is further supported by *Noraziah & Mohd Azlan (2012)* who stated that families who are poor role models influence the perception of the types of food that their children eat.

### Students' factors in choosing food

In choosing the food that they eat, two major factors contributed to the students' decision: cleanliness (65.8%) and the preference of their parents (12.3%). Cleanliness not only refers to the food but also to the way the food was prepared and the cleanliness of the premises where the food was bought. This finding shows that students are aware of the importance of cleanliness because contaminated food may cause food poisoning, cases of which have been reported in schools. As stated earlier, parents play an important role in determining the types of food their children eat. The results show that parents are aware of their children's eating habits and have influence on the types of food that their children eat. With the influence of parents, children have better guidance in the foods that they consume, which may prevent the consumption of unhealthy foods (*Cheng et al., 2008*).

### Students' perceptions of the ISHP and the implementation of health programs

According to the survey, only 45% of the students had heard of the ISHP that was implemented by the Ministry of Education. Furthermore, 28.2% of students had never heard of the program. This result was considered alarming because 40% of the respondents were in standard 6 and will soon graduate from primary school. Most of the students have been in primary school for five or six years but had never heard of the program, which may be due to a failure of the school administrators to clarify this program to the students. It is important that the students are always briefed and reminded to help them understand the importance of health and to incorporate awareness into the students' learning process. Failing to implement the program in schools deprives students of knowledge related to health issues. As such, *Basch (2010)* and *Egner, Oza-Frank & Cunningham (2014)* stated that in accordance to scientific reviews, schools that conduct health programs can create positive effects on students' educational outcomes and health outcomes. However, 77.8% of the students supported their school in events promoting healthy eating, which meant that the students had a positive attitude towards healthy eating events and showed an interest in participating in those activities. Nevertheless, due to a lack of clarity, 44.8% of the students were uncertain if they had joined the program in the previous year, and 29.2% stated that their schools had failed to conduct these programs. According to the results, it is important to show that instructions regarding any type of activities are very important. As the target students are in primary school, school administrators and teachers must play their roles in giving specific explanations and specific instructions in order for the students to clearly understand the purpose of conducting the programs. School administrators must also take the initiative of implementing more programs related to health. These programs should not be seen as a burden because active involvement from students will benefit them for years to come and will help the students in their future development. As the results show that students are supportive, school administrators should see this as a positive sign to help the students.

## LIMITATIONS

Certain limitations were found in this study. This study was limited to upper primary school students of standards 4, 5 and 6 and could be seen as not representing the whole population of primary school students. Furthermore, only 400 participants were chosen, and this study focused only on the state of Selangor. Moreover, the questionnaires were conducted in class under the supervision of teachers and not the representatives of the study. There is a possibility that the students provided answers that did not reflect their actual eating habits.

This study was only limited to student responses, but having responses from parents and the school administration would be more beneficial. With the additional responses of those two elements, the study could correlate the answers of the students, parents and administrators with respect to healthy eating. The data analysis was also limited to only frequencies and percentages.

Another major limitation is that all respondents may not have the similar ability to correctly recall their eating habits or some may over report their status. In questionnaire, the inclusion of grains (rice and noodles) in lunch box has been denoted as healthy. The term 'whole grains' would be more appropriate for further study as highly refined grains have similar effect as added sugars. Nonetheless, the study has provided important information on students' knowledge, attitudes and practices towards healthy eating and the influence of school programs in promoting healthy eating. The results will benefit future programs in helping students at the primary school level.

## CONCLUSIONS

The purpose of this study was to assess the knowledge, attitudes and practices of primary school students in healthy eating behaviors. It is concern that students have knowledge of the correct intake of foods but they still consume fried foods containing a high amount of saturated fat, cholesterol, sugar and salt. Most of the students are not having breakfast regularly that may cause serious health problem. Teachers and parents might give more attention to build up a habit of 'breakfast in time' among students. It was evident that school systems are lacking with regard to full implementation of their health programs. The ISHP was seen as a key element in inculcating a culture of health-conscious students. As students spend most of their time in school, fully utilizing the program is important to help students building healthy eating habit. Furthermore, students will be more committed with the help of teachers and friends. Students can also learn how to turn knowledge and theories into practice. In the students' responses regarding their own eating habits, the majority of students (74%) considered themselves healthy. By implementing health programs properly, all of the students' eating habits can be improved, thus creating a school with a healthy environment. The results of this study can be utilized by the Ministry of Education to look into improving the ISHP for the betterment of students learning a healthy lifestyle. The findings show that implementing more programs at the primary level to promote healthy eating can engage students in learning and practicing these healthy behaviors. Availability of healthier foods such as fruits, vegetables and whole grains in school campus can keep away students of changing the habits of having sugary drinks and fried foods. The school's cooperation with parents in discussing healthy snacks and meals for students to eat at school and at home can create a healthy environment at school.

### Funding

This project (RG300-14AFR) was funded by the Geran Penyelidikan Universiti Malaya (UMRG)–AFR (Frontier Science). The UM Research Cluster (Frontier Science) provided support in conducting this research. The funders had no role in study design, data collection and analysis, decision to publish, or preparation of the manuscript.

## Grant Disclosures

The following grant information was disclosed by the authors:

Geran Penyelidikan Universiti Malaya (UMRG)–AFR (Frontier Science): RG300-14AFR.

UM Research Cluster (Frontier Science).

## Competing Interests

There is no conflict of interest in any matter among authors.

## Author Contributions

- Kazi Enamul Hoque conceived and designed the experiments, wrote the paper, reviewed drafts of the paper.
- Megat Ahmad Kamaluddin performed the experiments.
- Ahmad Zabidi Abdul Razak prepared figures and/or tables.
- Afiq Athari Abdul Wahid analyzed the data, contributed reagents/materials/analysis tools, wrote the paper.

## Human Ethics

The following information was supplied relating to ethical approvals (i.e., approving body and any reference numbers):

The study protocol was approved by the University of Malaya Research Ethics Committee.

## Data Deposition

The raw data is included in the frequency distribution and percentage tables in the manuscript.

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
