# Peer review of "Building healthy eating habits in childhood: a study of the attitudes, knowledge and dietary habits of schoolchildren in Malaysia"

_PeerJ, doi:10.7717/peerj.2651_

## Round 0.1 · original submission · Minor Revisions

· Academic Editor

Minor Revisions

This article makes a valuable contribution to our understanding of dietary habits in children in the Asia Pacific region and it will be good to see the data published and available to all once the concerns addressed by the reviewers have been addressed. None of the points they have made are show stoppers, and most are relatively simply remedied. We look forward to seeing a revised version of the manuscript.

·

Basic reporting

• The introduction is well written to show context of the current issue.
• Please pay more attention on things such as spacing between paragraphs etc. It seems to vary throughout the paper.
• Please see the standard sections for PeerJ in the Instruction for authors (available at https://peerj.com/about/author-instructions/). Please revise your abstract according to the recommended headings (Background, Methods, Results, Discussion). I think this will improve clarity of your current abstract.
• Please also revise your main text according to the above Standard Sections (add Conclusion as one of the headings in your main text).
• Please revise the table numbering; currently your first table is Table 2. Did PeerJ format the current figure and table numbering?
• Introduction: please pay attention to the reference format that PeerJ uses, and please pay extra attention to the in-text citation. For example, in your first line of the Introduction (line 39):
Obesity has increased rapidly in incidence to become a global issue today (21, 22).
The first reference is reference number 21 and 22? This is somewhat confusing.
• Introduction: line 44: The information about non-communicable diseases (NCDs) is well known (superscript 1). I would suggest reducing the space spent on this.
• Introduction: please provide reference(s) for the information you provided in lines 47-49:
Furthermore, according to a study conducted by the Universiti Kebangsaan Malaysia (UKM), the percentage of obese children among primary school students in Malaysia has increased from 9.7% in 2001 to 13.7% in 2007.
• Introduction: please provide reference(s) for the information you provided in lines 57-67:
The existence of the Malaysian School Health Program began in 1967 with the main objective of developing students who are productive and healthy. In cooperation with the Ministry of Education and the Ministry of Health, the School Health Program was given the new name of the Program Bersepadu Sekolah Sihat (PBSS), or the Integrated School Health Program (ISHP). The purpose of the program was to synchronize all school activities that are related to health. Some activities that were introduced included the Rancangan Makanan Tambahan (RMT), or the Supplementary Food Program (SFP), which helps provide an adequate nutritional intake for students to ensure a well-balanced diet, and the Program Susu Sekolah (PSS), or the School Milk Program (SMP), which provides fresh milk for students to consume calcium. Other facilities, such as specific rooms for dental care and sick bays, were equipped in schools to aid in achieving the objectives of the ISHP.
• May I please ask the authors to provide us with the raw data, including a copy of the questionnaire?
• Please revise the titles of all tables and figures and include information on the study name or number of participants, country, and time. For example:
Table 2. Characteristics of 400 children aged 10-12 years in standard 4-6 in the state of Selangor, Malaysia 2015.
Figure 1. Students’ perception of their eating habits: data of 400 children aged 10-12 years in standard 4-6 in the state of Selangor, Malaysia 2015.
• Please check reference number 12. I did not seem to find that reference in the manuscript.

Experimental design

• This is an original primary research that falls within the scope of PeerJ and the authors clearly stated the knowledge gap that this study aims to fulfil.
• Methods: the information given about the study population is relatively limited. I only knew that the survey was conducted on children aged 10-12 years in grade 4-6 in the state of Selangor. There is no further information about the sampling frame, or the representativeness of the children/schools that were sampled. How many primary schools are there in the state of Selangor? This might be of interest to see whether the 16 schools (that were sampled) are representative of the state of Selangor. Please provide more information about this.
• Methods: given the consideration that ethical approval was obtained from the University of Malaya Ethics Committee, was the local Department of Education involved as well?
• Methods: the information provided in lines 85-87 might be better placed in the results section:
The majority of respondents (40%) were students in standard 6. The respondents also included more male students (53.5%) and more students belonging to the Malay race (65.8%).
• Methods: there are a few repetition in the text, for example:
Lines 82-83: The study was conducted among primary school students in standard four to standard six in the state of Selangor
Line 87: All respondents came from schools in the district of Selangor.
Another repetition:
Lines 70-72: The purpose of this study was to assess the knowledge, attitudes and practices of primary school students in Malaysia with respect to healthy eating behaviors.
Lines 88-89: an analysis was conducted to assess the knowledge, attitudes and practices of primary school students in the state of Selangor regarding healthy eating.
I would suggest reducing the space spent on this.
• Methods: please provide reference for the information you provided in line 113:
The questionnaire was adapted from the Hong Kong Department of Health (2009)
• Methods: please re-arrange the procedure chronologically.
• Methods: please provide more information on the data analysis section.

Validity of the findings

• Overall, the findings are of interest. There is perhaps some room for improvements in the way findings are presented; including a more thorough checking about the information presented in the tables and figure.
• Results: the use of subheadings is appropriate and really helpful for readers to follow the flow of the papers, however, I think the manuscript would benefited by presenting the results in accordance with the aim of the study: knowledge, attitude, and practices (in that order).
• Results: please provide more information about Table 2. The previous information deleted from the Methods section would fit nicely here:
The majority of respondents (40%) were students in standard 6. The respondents also included more male students (53.5%) and more students belonging to the Malay race (65.8%).
• Results: please consider to do a thorough checking about which information should be presented in the Methods section, and which one belongs to the Results section, for example in lines 145-146:
In identifying knowledge about healthy food, six pairs of food were given in the questionnaire. The students were asked to choose the healthiest food among the six pairs.
Another example:
In identifying the students’ food preferences, the same six pairs of food and drink as the knowledge questions were provided.
Another example:
Students were shown pictures of snacks before answering whether they had the habit of eating them.
Supplementary figure/file that shows the pairs of food/complete questionnaire might be helpful.
• Results: please check the information you provided in Table 3, comparison between soya sauce and deep fried drumsticks. Do you mean chicken cooked with soya sauce? I do not think soya sauce is comparable to deep fried drumsticks. Please explain if this is a cultural thing in Malaysia to eat soya sauce solely.
• Results: please check lines 159-165, these two paragraphs are overlapping and quite confusing. The main reason why students did not have breakfast was not available in Table 3.
Students’ habits of consuming breakfast
A total of 67%, or 268 students, had consumed breakfast on the day of the survey. The main reason why 33% of the students did not have breakfast was due to inadequate time (56.8%), as shown in Table 3.
Reasons for not consuming breakfast
Table 4 shows that an alarming figure of 56.8% of the students did not have breakfast because they did not have enough time, and 22.7% were not in the habit of having breakfast.
• Results: lines 173-174:
Between 60% and 92.7% of the students did not reach the recommended frequencies, as shown in Table 6.
Please check this sentence, I cannot seem to translate Table 6 into the above sentence. Recommended frequencies of what in particular? If in general, it should be between 19.5%-92.7%.
• Results: please check Table 6, the authors have duplication of “not having food high in salt” (last 2 lines).
• Results: lines 180-183:
Based on the two questions asked, there was a lack of healthy eating knowledge by the students because only 27.5% correctly chose grains as the type of food that should be provided most often in a healthy lunchbox (24), and only 35.8% knew the correct recommended servings of fruits and vegetables (Table 7).
The Results section should provide findings from the current study; any comparison with other study should be placed in the Discussion section.
• Discussion: it would be better to highlight the most important finding in the beginning of the first paragraph
• Discussion: please consider re-arranging the Discussion section to be in line with the results section so it will be easier for readers to follow the flow of the paper.
• Some of the information stated to support the findings were not well referenced. If these were based on speculations, the authors should identify those sentences as such.
• Please provide conclusion that clearly responded to the aim of the study.

Additional comments

This study has chosen a significant topic. Overall, some of the issues highlighted in the study are of great importance. However, I would suggest a few improvements to be addressed before the paper is suitable for publication, including some tight editing and some clearer layout. In particular, more complete and well-structured information on methods, results, and discussion are needed in order to aid interpretation of the results. Specific comments about the papers are included.

Reviewer 2 ·

Basic reporting

There is a lack of clarify in expression throughout the manuscript and can be improved by using professional English language services or checked by native English speaker. The references used in the introduction chapter need to be updated especially in reference to the statistics of childhood obesity. The explanation on the school health programme was sufficient. However, there is a lack of link between the explanation and the lack of awareness on health programme to explain the gap in the literature, hence the need of the study. There are multiple tables in the results section that can be combined while some data can be more appropriately expressed in figures.

Experimental design

The purpose of the study is clear and ethics approval was reported. However, despite being piloted, the study instrument was insufficiently described, creating much confusion in the results section. The explanation of methods can be improved with the use of figures to illustrate the types of schools and the flow of study procedure. The methods used to perform data analysis was not mentioned.

Validity of the findings

There is an overlapping/ repetition of results mentioned in the discussion section. Much of the discussion section was not well-referenced to the current literature on dietary habits and health issues of school children. The style of writing demonstrated lack of cohesiveness and mostly authors’ own speculation.

Additional comments

Not Applicable

Reviewer 3 ·

Basic reporting

Reporting was clear.

Experimental design

Major limitation was the self-reported dietary intake, which is likely subjected to systematic recall bias (i.e. students over reporting healthier habits) as well as large random error (i.e. inability to correctly recall eating habits). These limitations should be acknowledged in the discussion.

Authors should also provide descriptive statistics for the types of schools where the data was collected from to enable interpretation of generalizability.

There were some other issues with the design of the study - for example, the choice of paired foods were somewhat problematic Soy sauce and deep fried foods are not comparable products. It is also debatable whether soy sauces loaded with sodium could be considered a healthy option.

Validity of the findings

Reasonable.

Additional comments

Updated dietary guidelines (e.g. see the most recent USA Dietary Guidelines Advisory Committee report) now acknowledge that dietary cholesterol is not a nutrient of concern given the lack of material association between dietary cholesterol and serum cholesterol or clinical cardiovascular disease in general populations. Similarly, not all dietary fats are harmful - strong evidence clearly suggest that substitution of saturated fat with polyunsaturated fat will improve cardimetabolic outcomes (e.g. see http://journals.plos.org/plosmedicine/article?id=10.1371/journal.pmed.1002087). Finally - to state that grains should be the type of food that should be most commonly provided in school lunch boxes is simply incorrect. Highly refined grains have similar physiologic effect as added sugars, and dietary emphasis should be on having whole grains, not just 'grains'.

Please revised the manuscript throughout to reflect these new evidence when discussing what should be considered 'healthy foods and nutrients', e.g. by taking out references to cholesterol in foods, and use 'saturated fat' in place of 'fat' to add nuance to discussions. Authors should also acknowledge these deficiencies in their survey instrument (e.g. referring to just grains), which should be updated in future studies to ensure they reflect the latest evidence of nutrition science.

Authors should add brief discussion on other types of school nutrition programs that require healthier food provision (e.g. bans on sugary drinks and ensuring the majority of foods available are 'healthier' core foods like fruits, vegetables, whole grains) as a way to improve the school food environment.

---

## Round 0.2 · accepted · Accept

· Academic Editor

Accept

Your revision has addressed the key issues raised by the reviewers and your paper is now in a reasonable state to be published.